# Neuroblastoma Cells Depend on CSB for Faithful Execution of Cytokinesis and Survival

**DOI:** 10.3390/ijms221810070

**Published:** 2021-09-17

**Authors:** Elena Paccosi, Michele Costantino, Alessio Balzerano, Silvia Filippi, Stefano Brancorsini, Luca Proietti-De-Santis

**Affiliations:** 1Unit of Molecular Genetics of Aging, Department of Ecology and Biology, University of Tuscia, 01100 Viterbo, Italy; e.paccosi@unitus.it (E.P.); Cosmic21@hotmail.it (M.C.); alessio.balzerano@unitus.it (A.B.); silvia.filippi@unitus.it (S.F.); 2Unit of Molecular Pathology, Department of Experimental Medicine, Section of Terni, University of Perugia, 06100 Perugia, Italy; stefano.brancorsini@unipg.it

**Keywords:** neuroblastoma, cytokinesis failure, cell death, gene therapy, Cockayne Syndrome

## Abstract

Neuroblastoma, the most common extra-cranial solid tumor of early childhood, is one of the major therapeutic challenges in child oncology: it is highly heterogenic at a genetic, biological, and clinical level. The high-risk cases have one of the least favorable outcomes amongst pediatric tumors, and the mortality rate is still high, regardless of the use of intensive multimodality therapies. Here, we observed that neuroblastoma cells display an increased expression of Cockayne Syndrome group B (CSB), a pleiotropic protein involved in multiple functions such as DNA repair, transcription, mitochondrial homeostasis, and cell division, and were recently found to confer cell robustness when they are up-regulated. In this study, we demonstrated that RNAi-mediated suppression of CSB drastically impairs tumorigenicity of neuroblastoma cells by hampering their proliferative, clonogenic, and invasive capabilities. In particular, we observed that CSB ablation induces cytokinesis failure, leading to caspases 9 and 3 activation and, subsequently, to massive apoptotic cell death. Worthy of note, a new frontier in cancer treatment, already proved to be successful, is cytokinesis-failure-induced cell death. In this context, CSB ablation seems to be a new and promising anticancer strategy for neuroblastoma therapy.

## 1. Introduction

Neuroblastoma is an embryonic neoplasm of the sympathetic nervous system arising from the neuronal precursor cells of the sympathoadrenal lineage of the neural crest [1,2]. It is the most common extra-cranial solid tumor of early childhood [3]. Neuroblastoma is a complex disease that has high genetic, biological, clinical, and morphological heterogeneity, so the treatment in pediatric populations is complex and difficult, and there is a high incidence of death, regardless of the use of intensive multimodality therapies [4,5,6].

It is widely accepted among molecular oncologists that, despite the array of qualitative and quantitative genetic alterations typical of cancer, some tumors are addicted to a single gene for survival and proliferation [7]. As a consequence, the inhibition of this specific gene is often sufficient to halt the neoplastic process. Oncogenic factors are potentially good therapeutic targets since they often have increased expression and activity in a variety of cancers. CSB, a member of the SWI/SNF ATP-dependent chromatin remodeling complex [8,9], whose loss of function is known to be responsible for a premature aging syndrome, called Cockayne’s syndrome, was recently found to be over-expressed in a variety of cancer tissues and cells [10]. We demonstrated that a number of cancer cell lines of different tissue origins display a dramatic up-regulation of CSB protein expression and are addicted to these increased levels of CSB [11]. Indeed, upon CSB suppression in these cells, several pro-apoptotic factors become dramatically up-regulated, leading to a massive induction of apoptosis [12]. Moreover, CSB ablation specifically affects tumor cells without harming non-transformed cells [11], suggesting that only the former are addicted to high levels of CSB. Worthy of note, abolishment of CSB has also shown anticancer effects in vivo: disruption of the *CSB* gene reduces the tumor rate in cancer-predisposed INK4a/ARF −/− mice [13]. At a molecular level, CSB protein exhibits ATPase activity and has conserved helicase motifs [9,14,15]. CSB was first characterized as a DNA repair protein, playing a role in the transcription-coupled repair (TCR), a sub-pathway of Nucleotide Excision Repair (NER) devoted to the rapid removal of the transcription-blocking lesions located on the coding strands [16,17,18]. However, more recent findings indicated that CSB is a multifaceted protein implicated not only in DNA repair but also in gene expression modulation and cell division [19,20,21,22,23]. Its function in the ubiquitination/degradation mechanism presumably underlies its role in all the above processes [24]. Therefore, CSB involvement in the context of DNA repair, transcription, and cell division, as well as in the modulation of critical pathways such as the p53 [25], the hypoxia [26], and the UPR responses [12], suggest that cancer cells might benefit from its overexpression. In this study, we demonstrated that the tumorigenicity of neuroblastoma cells is reduced after CSB suppression. In fact, we found that CSB depletion caused cytokinesis failure, which decreased cell proliferation and triggered apoptosis via activation of executioner caspases 9 and 3. These in vitro results open a new and promising scenario for neuroblastoma therapy, in which CSB targeting, if validated in vivo, could be a potential and safe possibility.

## 2. Results

### 2.1. CSB Suppression Affects Proliferation, Clonogenicity, and Invasivity of Neuroblastoma Cells

Western blot analysis of CSB protein amount showed that neuroblastoma cells IMR-32 and SK-N-BE (2c) over-express CSB protein compared to normal neural cells (RenVM) (Figure 1A,B). To evaluate the effect of CSB suppression in the neuroblastoma cell line SK-N-BE (2c), which displays the upper levels of CSB expression, a lentiviral negative control vector containing scrambled shRNA and a lentiviral-positive vector containing shRNA, specifically targeting *CSB*-mRNA, were used to create a cell line that maintains CSB proficiency (sh-K) and a cell line in which CSB is suppressed (sh-CSB) (Figure 1C).

Stably transduced cells were selected by puromycin (2 μg/mL) resistance. Both Western blot analyses (Figure 1D) and RT-PCR of puromycin-resistant SK-N-BE (2c) cells showed a good suppression of CSB expression (73%) in the sh-CSB cell line relative to the scrambled vector-transfected cells (Figure 1E).

The impact of CSB suppression on the cells’ viability and proliferation was analyzed with an MTT [3-(4,5-dimethylthiazol-2-yl)-2,5-diphenyl-2H-tetrazolium bromide] assay. We observed a significant difference between CSB-proficient and suppressed cells with a dramatic decrease of cell viability in sh-CSB cells across the entire time course of analysis (Figure 2A). We also performed a long-term proliferation assay counting the number of cells every 3 days. This assay confirmed that CSB suppression affects the ability of cells to retain their reproductive integrity over a prolonged period of time: sh-CSB cells did not show an exponential growth rate compared with sh-K cells that, instead, reached a plateau after 12 days from seeding (Figure 2B). In order to test whether sh-CSB cells retain colony formation capacity and adherence to substrate, a clonogenic assay was performed seeding cells at low density and maintaining them in culture for at least 15 days until colony formation. After fixation and staining, the count of colonies showed a significantly different capacity of colony formation between sh-K and sh-CSB cells, suggesting a decrease in clonogenicity of neuroblastoma cells after CSB suppression (Figure 2C,D).

We next tested the invasivity of sh-K and sh-CSB cells by a Scratch cell migration assay, observing that the lack of expression of CSB protein drastically reduced the recovery of a wound in neuroblastoma cancer cells across the 48 h time course analysis (Figure 2E,F).

### 2.2. CSB Suppression Determines a Massive Apoptosis Induction in Neuroblastoma Cells

Finally, we quantitatively assessed the rates of apoptosis in SK-N-BE (2c) sh-K and sh-CSB cells by using a combination of fluorescent dyes to analyze the morphological alterations that cells undergo during apoptosis: fluorescein diacetate (FDA) and Hoechst (HO) stains the cytoplasm and the nucleus, respectively, of viable cells, while the necrotic and the late-stage apoptotic cells were stained by propidium iodide (PI), which requires the loss of the cytoplasm and nuclear membrane integrity. Our results showed a largely significant increased rate of apoptosis in sh-CSB cells (Figure 3A panel b and Figure 3B) as compared to sh-K cells (Figure 3A panel a and Figure 3B). To verify if CSB suppression selectively induces apoptosis in cancer cells rather than normal cells, we performed the same experiment in RenVM sh-K and sh-CSB cells. Noteworthy, CSB suppression in RenVM cells does not induce a significant increase of the apoptotic rate compared to the control cells (Figure 3A panels c and d and Figure 3B). Next, by confocal microscopy, we checked the activation of caspase 9 and 3, unequivocal apoptosis markers [27], by using antibodies that specifically recognized the cleaved forms. We observed a drastic induction of caspases cleavage only in CSB suppressed cancer (SK-N-BE (2C)) cells, but not in normal (RenVM) cells, thus confirming cancer-selective apoptosis triggering (Figure 3C–G). Western blot analysis confirmed the massive activation of caspase 3 after CSB suppression only in SK-N-BE (2C) cells (Figure 3H,I).

### 2.3. Centrosomal Abnormalities and Spindle Multipolarity in CSB Suppressed Neuroblastoma Cells

The morphological differences between sh-K and sh-CSB SK-N-BE (2c) cells were very significant. Immunostaining against cytoskeletal proteins revealed that the shape of sh-K cells appears normal (Figure 4A, panels a–c), while the morphology of CSB-suppressed cells was similar to that of replicative senescent cells, with enlarged shape (Figure 4A, panels d–f), abnormal nuclei, and a high frequency of multinucleation (82.3% ± 3.4% *n* = 500; Figure 4A, panels g–i).

Furthermore, CSB-suppressed cells showed a very high frequency of aberrant mitosis, characterized by the presence of spindle multipolarity (85.5% ± 4.5%, n = 500; Figure 4B, panels d–f). These abnormalities were not displayed by sh-K cells (3.03% ± 1.5%, n = 500; Figure 4B, panels a–c). In contrast, RenVM sh-CSB cells did not show a drastic increase in multinucleation, since we observed only a few binucleated cells (3% ± 0.6%, n = 500) compared to sh-K control cells (0.8% ± 0.5%, *n* = 500) (Appendix A, panels a and b). Furthermore, both RenVM sh-K and sh-CSB cells did not display abnormal multipolar spindles (Appendix A, panels a–f).

Staining of mitotic cells with anti-gamma-tubulin antibody revealed that in SK-N-BE (2c) sh-K cells, the two centrosomes correctly form the poles of a bipolar mitotic spindle, while SK-N-BE (2c) sh-CSB cells mitotic cells displayed multipolar spindles characterized by supernumerary centrosomes (Figure 5A, panels a and b, and Figure 5B). Noteworthy, in interphase, supernumerary centrosomes localized in a central position, surrounded by radially disposed nuclei (Figure 5C panels c–d). The phenotype that we observed is described as typical of the polyploidization stage, with a radial arrangement of genomes and a central MTOC (MicroTubules Organizing Center) [28].

It is clear that spindle multipolarity seen in sh-CSB cells during mitosis arises from this centrosome’s surplus, undetectable in sh-K cells (Figure 5B panels a–b), and that is due to the cytokinesis failure described by Paccosi and colleagues in cells defective in CSB expression [23]. Accordingly, similar phenotypes were displayed by sh-K after cytochalasin-induced cytokinesis failure, with the appearance of radially disposed nuclei (Figure 5D, panels a–c) and multipolar spindles (Figure 5D, panels d–f).

## 3. Discussion

The discovery and validation of new targets for treatments of cancers are important for improving patient survival. The identification of new therapeutic options is particularly required to fight neuroblastoma, being that chemotherapy, radiotherapy, and surgical methods demonstrated low efficacy, particularly in the late stages of the treatment of this disease [29,30,31]. In this context, cell death-based strategies have recently been shown to be successful for neuroblastoma treatment [32].

A previous study of our group showed that a number of cancer cell lines from different tissues, including bladder, breast, cervix, and prostate, display a dramatically increased expression of the CSB protein, a DNA repair factor that has recently been shown to be involved in cell robustness [11]. Furthermore, ablation of this protein causes devastating effects on tumor cells through a drastic reduction of cell proliferation and massive induction of apoptosis. In this state of the art, we wondered if suppression of CSB might also reduce neuroblastoma cells tumorigenicity. In fact, we demonstrated that neuroblastoma cells, after CSB suppression, display multinucleation, supernumerary centrosomes, and spindle multipolarity following a long-term reduction of cell proliferation and massive induction of apoptosis. Indeed, as shown in Figure 3C, panel b and 3E, panels b–d, multinucleated cells undergo apoptosis via activation of caspase 9 and 3, while CSB suppression in normal neural cells does not trigger apoptosis (Figure 3D panel b and 3F panel b).

In Hela cells, the massive induction of apoptosis due to CSB suppression correlates with a prolonged Unfolded Protein Response (UPR) that induces ER stress-associated programmed cell death [12]. In neuroblastoma cells, CSB suppression correlates with cellular phenotypes attributable to cytokines failure, followed by the induction of cell death. This phenomenon is likely linked to the role that CSB, together with its partner CSA, plays in cell division. Indeed, we recently unveiled that CSB and CSA are responsible for the triggering of cytokinetic abscission by promoting the ubiquitination and the degradation of PRC1. We showed that defects in CSA or CSB result in perturbation of the abscission, leading to the formation of long intercellular bridges and multinucleated cells [23]. The existence of two different mechanisms that lead to the death of cancer cells in the absence of CSB is compatible with its pleiotropic function. What is at the basis of the triggering of one of these two alternative ways is not clear yet. Some works suggest that polyploidy and Bcl2 and/or p53 status may be involved in determining which of these pathways will be promoted. For Bcl2, it was demonstrated that cancer cells expressing lower levels of this anti-apoptotic protein are more sensitive to cytokinesis-failure-induced apoptosis [33,34,35], while, regarding p53, its contribution in determining the cellular response following polyploidization is still under debate due to p53 multiple roles [36]. Regardless, several reports suggest that p53 proficiency or mutational status correlated with resistance and sensitivity to apoptosis, respectively [37,38,39].

Worthy of note, growing evidence seems to suggest that a useful approach for cancer treatment might be to increase polyploidy of cancer cells by interfering with cell division above a threshold that would be incompatible with cell viability [40,41,42]. Along this line, previous works showed that a reduced expression of proteins involved in cytokinesis, such as Aurora kinases and Plk, are associated with low levels of cytokinesis failure that selectively affect rapidly dividing cells, such as cancer cells, without harming organ growth and functioning [43,44,45]. Such a strategy seems to work well in neuroblastoma. Therefore, this might suggest that *CSB* silencing would selectively affect the proliferation and survival of cancer cells without harming normal tissue and organ functionality. Accordingly, growth failure and neurodegeneration have been described in humans only when the function of the *CSB* gene product is completely lost, as in the case of homozygous loss of function mutations [46,47]. Of course, it is important to take into account the risk that, on the other side, cytokinesis failure and polyploidy might also promote tumorigenesis, chromosome instability (CIN), and drug resistance [48,49,50,51]. Furthermore, in our experience, the fate of polyploid and multinucleated neuroblastoma cells, induced by CSB ablation, is death, as shown by the large induction of executioner caspases 3 and 9.

Worthy of note, therapeutics inducing cell death, mainly apoptosis, have been proven to be successful in neuroblastoma treatment [32]. Until now, different approaches to cell death have been applied, targeting p53/MDM2 interaction [52], changing the balance between pro- and anti-apoptotic proteins such as Bcl-2 and Mcl-1 [53], and targeting the PI3K/AKT/mTOR pathway [54].

Unfortunately, the main problem with this kind of strategy is the ability of tumor cells to compensate for pro-apoptotic signals via up-regulating the anti-apoptotic ones. Therefore, searching for new strategies is mandatory in order to achieve an improved outcome for neuroblastoma therapy. In this scenario, apoptotic cell death induction via CSB ablation could be a new and promising anticancer strategy.

## 4. Material and Methods

### 4.1. Cell Culture and Silencing

SK-N-BE (2c) cells (American Type Culture Collection (ATCC) CRL-2271) were grown in a MEM/DMEM F12 medium supplemented with 10% Fetal Bovine Serum and 2 mM l-Glutamine.

IMR-32 cells (American Type Culture Collection (ATCC) CCL-127) were grown in a MEM medium supplemented with 10% Fetal Bovine Serum, 2 mM L-Glutamine, and Non-Essential Amino Acids.

ReNCell VM cells (SCC008, Sigma-Aldrich: St. Louis, MO, USA) were grown as an adherent monolayer on poly-ornithine (0.002%) laminin (2 mg/mL) coated tissue culture flasks in the presence of 20 ng/mL of human recombinant EGF and bFGF2 in DMEM:F12 medium with nutrients optimized for neural progenitor cell growth.

Cells in exponential growth phase were transduced with lentiviral shRNA particle (1 × 105 infectious units of virus—Santa Cruz Biotechnology, Santa Cruz, CA, USA), expressing sh-RNA targeting *CSB* or sh-RNA non-targeting control. Puromicin selection (2 μg/mL) is performed to achieve stable gene silencing.

Cells were treated with 5 μg/mL of Cytochalasin B for 24 h.

### 4.2. Western Blot Analysis

Proteins were fractionated by sodium dodecyl sulfate/poly-acrylamide gel electrophoresis (SDS/PAGE) and transferred to Protran nitrocellulose membranes (Sigma) and blotted with respective antibodies.

### 4.3. Retrotranscription and Real-Time Quantitative PCR

RNA was isolated using the NucleoSpin RNA II kit (Macherey-Nagel GmBH & Co., Dueren, Germany). cDNA synthesis was performed using the First Strand cDNA Synthesis kit (Fermentas, St. Leon-Rot, Germany). Real-time quantitative PCR was carried out with SYBR green master mixture (Promega, Madison, WI, USA) using Mx3005P Real-Time PCR system (Agilent, Santa Clara, CA, USA). Results were normalized to β-actin. Primers sequences are available upon request.

### 4.4. MTT [3-(4,5-Dimethylthiazol-2-yl)-2,5-diphenyl-2H-tetrazolium Bromide] Cell Proliferation Assay

Cells were plated in 12-well plates, and MTT was added to each well (0.5 mg/mL) at the indicated times. After incubation for 3 h at 37 °C, the supernatant was replaced with 750 µL of solution (10% SDS, 0.6% acetic acid in DMSO) to dissolve the formazan crystals and produce a purple solution. After the split in a 96-well plate of 150 µL of each sample in quadruplicate, optical density measurements were obtained using a scanning spectrophotometer DTX 880 Multimode Detector (Beckman Coulter, Brea, CA, USA). The readings were made using a 630 nm (background) and a 570 nm filter.

### 4.5. Long-Term Proliferation Assay

To assay differences in cellular proliferation between control and suppressed cell lines, 200,000 cells for each line were seeded in as many flasks as there were experimental times; then, every 3 days, one flask for each cell line was trypsinized, and cells were counted using the Nucleo counter (Chemometec, Allerod, Denmark).

### 4.6. Clonogenic Assay

Clonogenic assay was performed through seeding in each well of 6-well plate cells at low density (500 cells/well). Plates were grown at 37 °C in a humidified atmosphere of 5% CO2/95% air for at least 15 days. When colony number differences were evident, cells were fixed and stained with a mixture of 6.0% glutaraldehyde and 0.5% crystal violet and counted.

### 4.7. Scratch Cell Migration Assay

Motility and invasivity of sh-K and sh-CSB cells were assessed by the scratch cell migration assay. Cells were seeded in 60 mm plates. When 90% confluence was achieved, a 20–200 uL pipette tip was used to “scratch” the bottom of the plate. Photographs of the “wound healing” were taken at 0, 24, and 48 h, and the length of the scratch was measured with the AxioVision Rel. 4.8 program (Carl Zeiss Microscopy GmbH, Jena, Germany).

### 4.8. Apoptosis Assay

A combination of fluorescein diacetate (FDA; 15 μg/mL), propidium iodide (PI, 5 μg/mL), and Hoechst (HO, 2 μg/mL) was used to differentiate apoptotic cells from viable cells. FDA and HO are vital dyes that stain the cytoplasm and nucleus of the viable cells, respectively. The necrotic and the late stage of apoptotic cells are readily identified by PI staining. Approximately 500 randomly chosen cells were microscopically analyzed to determine apoptosis levels.

### 4.9. Immunofluorescence

For immunofluorescence experiments, cells were seeded onto Ibidi coverslips. Cells were fixed in ice-cold methanol or 2% formaldehyde, washed three times in PBS, permeabilized in 0.25% Triton X- 100, in PBS for 10 min, and then blocked in 3% bovine serum albumin in PBS for 30 min before the required primary Abs were applied. The following Abs were employed: anti-alpha-tubulin moAb (Sigma, Steinheim, Germany), rabbit anti-gamma-tubulin Ab (Sigma, Steinheim, Germany), anti-B-actin moAb (Santa Cruz Biotechnology, Santa Cruz, CA, USA), rabbit anti-CSB (N2C1), Internal Ab (GeneTexm Alton Pkwy Irvine, CA, USA), rabbit anti-cleaved caspase-9 (Cell Signaling Technology, Danvers, MA, USA), and rabbit anti-cleaved caspase-3 (Cell Signaling Technology, Danvers, MA, USA). Appropriate secondary Alexa Fluor Abs (Thermo Fisher Scientific, Waltham, MA, USA) were used. DNA was marked with 4′,6-diamidino-2-phenylindole (DAPI) in Vectashield. Slides were analyzed with a confocal microscope system (Zeiss LSM 710, Carl Zeiss Microscopy GmbH, Jena, Germany), and images were acquired using the interfaced software ZEN 2010. Both microscope hardware and software configuration were always maintained. Technical parameters fixed in our acquisition procedure were pinhole size, at 1 AU (Airy unit), laser power at 2%, and digital gain at 1.0. Images were then processed using ImageJ software (http://rsbweb.nih.gov/ij/) in order to merge channels from monochrome acquisitions and make montage, when serial microscope scans of the specimen were performed along z axis.

## Figures and Tables

**Figure 1 ijms-22-10070-f001:**
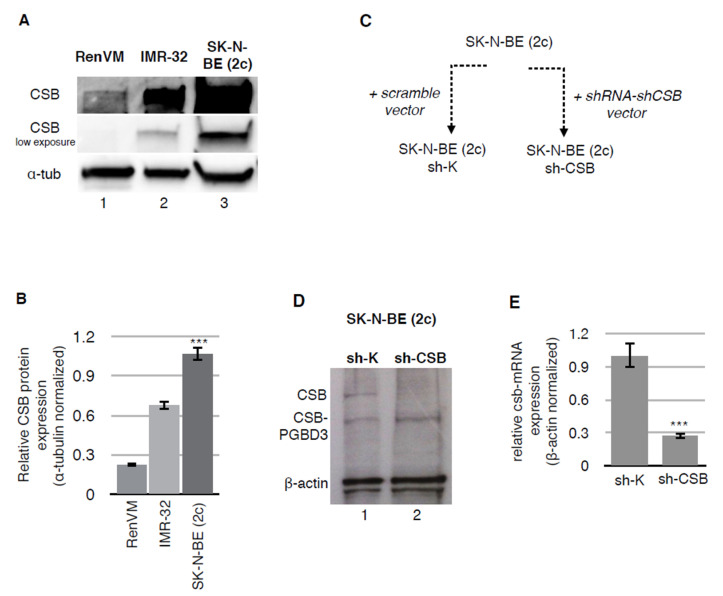
CSB is overexpressed in neuroblastoma cell lines. CSB protein expression levels in RenVM, IMR-32, and SK-N-BE (2c) cellular extracts. Western blot has been performed using antibodies against CSB (with lower exposure for saturated signal) and α-Tubulin (**A**). Graph showing relative CSB expression α-Tubulin normalized from Western blot in panel A (**B**). Schematic representation of *CSB* silencing in SK-N-BE (2c) cells (**C**). CSB protein expression levels in SK-N-BE (2c) sh-K and sh-CSB cells. Western blot has been performed using antibodies against CSB and β-actin. CSB-PGBD3 indicates the fusion protein generated by the integration of *piggyBac3* transposable element into intron 5 of the *CSB* gene (**D**). Graph showing relative *CSB*-mRNA expression β-actin normalized in SK-N-BE (2c) sh-K and sh-CSB cells (**E**). Each Western blot is the representation of three independent biological repeats. ***: *p* < 0.001.

**Figure 2 ijms-22-10070-f002:**
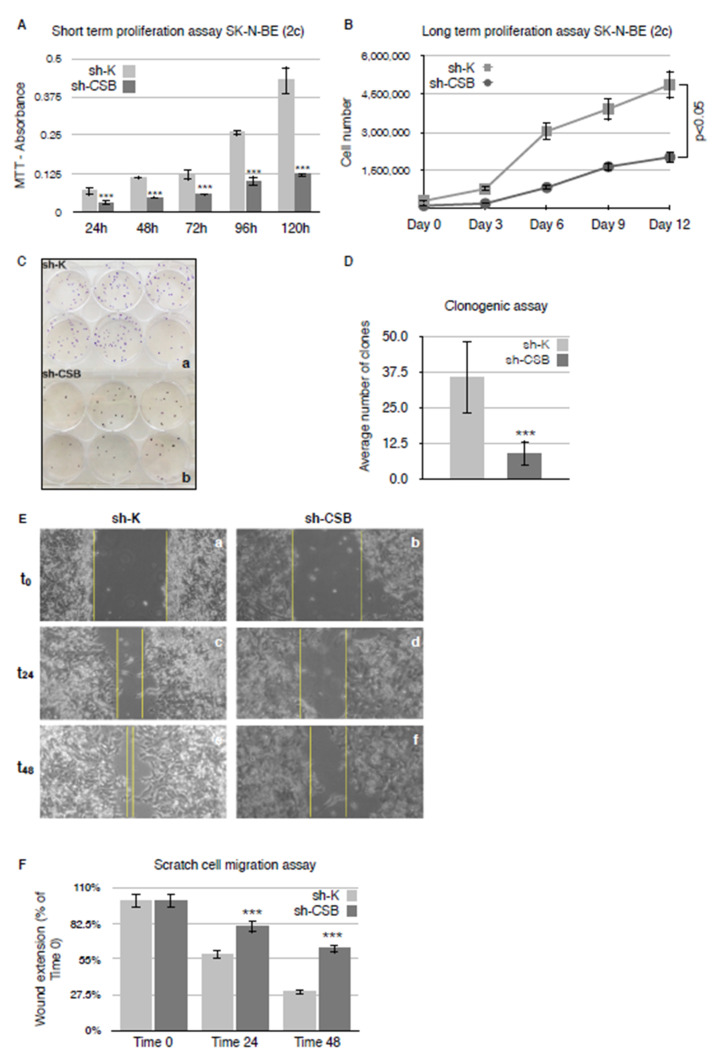
CSB suppression affects proliferation, clonogenicity, and invasivity of neuroblastoma cells. Graph showing absorbance of MTT cell proliferation assay from SK-N-BE (2c) sh-K and sh-CSB cells at different time points from seeding (**A**). Graph of long-term proliferation assay showing SK-N-BE (2c) sh-K and sh-CSB cell numbers at different time points from seeding (**B**). Clonogenic assay on SK-N-BE (2c) sh-K and sh-CSB cells (**C**). Graph showing the average number of SK-N-BE (2c) sh-K and sh-CSB cell clones in clonogenic assay (**D**). Micrographs at different time points of scratch cell migration assay performed on SK-N-BE (2c) sh-K and sh-CSB cells (**E**). Graph showing the percentage of wound extension from Time 0 in SK-N-BE (2c) sh-K and sh-CSB cells at different time points (**F**). Values are from three independent biological repeats (mean ± S.D., one-way ANOVA with Tukey’s post hoc test was used for statistical analysis). ***: *p* < 0.001.

**Figure 3 ijms-22-10070-f003:**
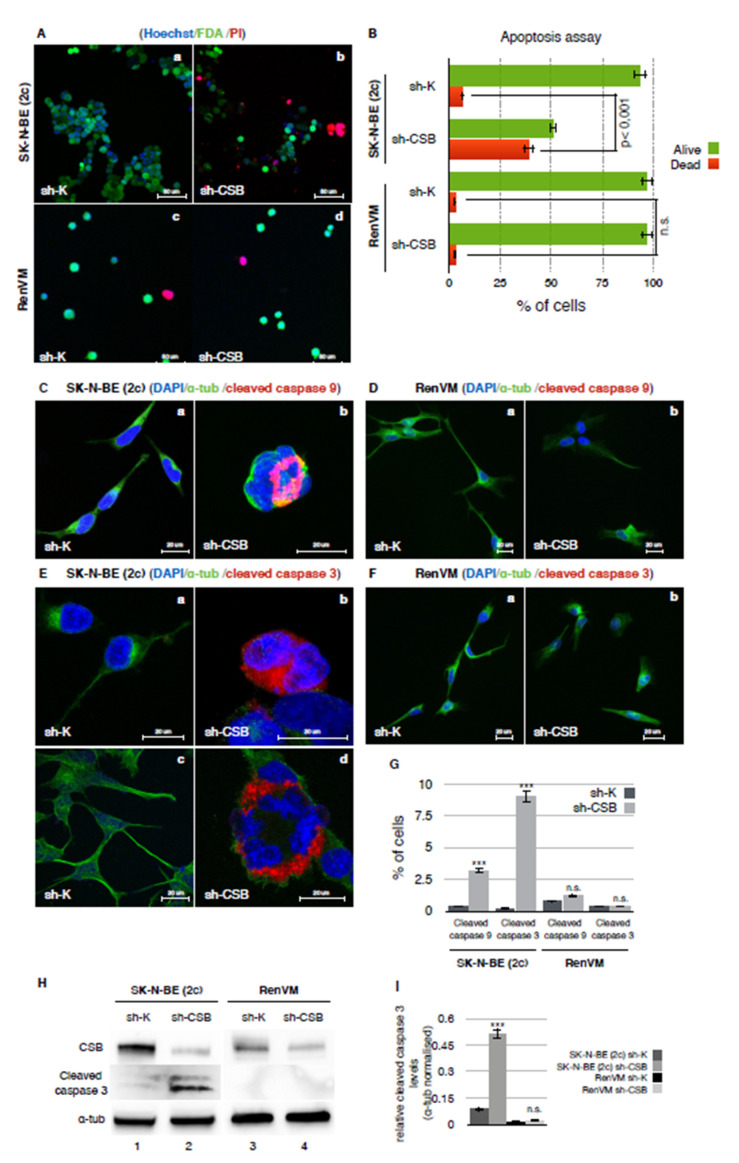
CSB suppression determines a massive apoptosis induction in neuroblastoma cells. Confocal micrographs of SK-N-BE (2c) sh-K (panel a), sh-CSB (panel b), RenVM sh-K (panel c), and sh-CSB (panel d) cells stained for apoptosis assay with Hoechst (blue), fluorescein diacetate (FDA, green), and propidium iodide (PI, red) (**A**). Graph showing the percentage of alive (green) and dead (red) cells from apoptosis assay (*n* = 500 × 3 independent experiments) (**B**). Confocal micrographs of SK-N-BE (2c) sh-K (panel a) and sh-CSB (panel b) cells, stained for DNA (blue), α-tubulin (green), and cleaved caspase 9 (red) (**C**). Confocal micrographs of RenVM sh-K (panel a) and sh-CSB (panel b) cells, stained for DNA (blue), α-tubulin (green), and cleaved caspase 9 (red) (**D**). Confocal micrographs of SK-N-BE (2c) sh-K (panel a and c) and sh-CSB (panel b and d) cells, stained for DNA (blue), α-tubulin (green), and cleaved caspase 3 (red) (**E**). Confocal micrographs of RenVM sh-K (panel a) and sh-CSB (panel b) cells, stained for DNA (blue), α-tubulin (green), and cleaved caspase 3 (red) (**F**). Graph showing the percentage of SK-N-BE (2c) sh-K and sh-CSB, RenVM sh-K, and sh-CSB cells stained for cleaved caspase 9 and 3 (n = 100 × 3 independent experiments) (**G**). Cleaved caspase 3 protein expression levels in SK-N-BE (2c) sh-K, sh-CSB, RenVM sh-K, and sh-CSB cellular extracts. Western blot has been performed using antibodies against CSB, cleaved caspase 3, and α-tubulin, and is the representation of three independent biological repeats (**H**). Graph showing levels of cleaved caspase 3, α-tubulin normalized in SK-N-BE (2c) sh-K, sh-CSB, and RenVM sh-K and sh-CSB cells (**I**). N.s.: not significant. ***: *p* < 0.001.

**Figure 4 ijms-22-10070-f004:**
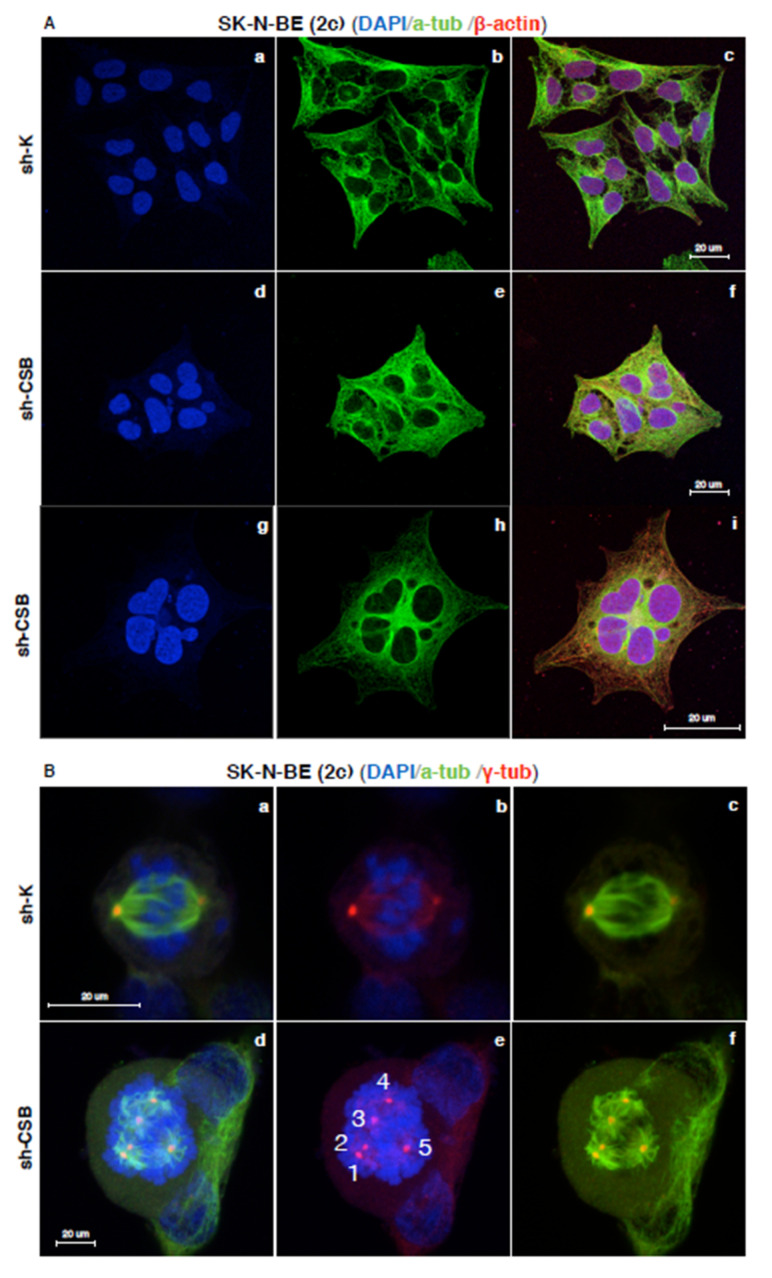
CSB suppression gives rise to multinucleated cells and spindle multipolarity. Confocal micrographs of SK-N-BE (2c) sh-K (panels a–c) and sh-CSB (panels d–i) cells stained for DNA (blue), α-tubulin (α-tub, green), and β-actin (red) (**A**). Confocal micrographs of SK-N-BE (2c) sh-K (panels a–c) and sh-CSB (panels d–f) cells stained for DNA (blue), α-tubulin (α-tub, green), and γ-tubulin (γ-tub, red). Supernumerary centrosomes are indicated by white numbers (panel e) (**B**).

**Figure 5 ijms-22-10070-f005:**
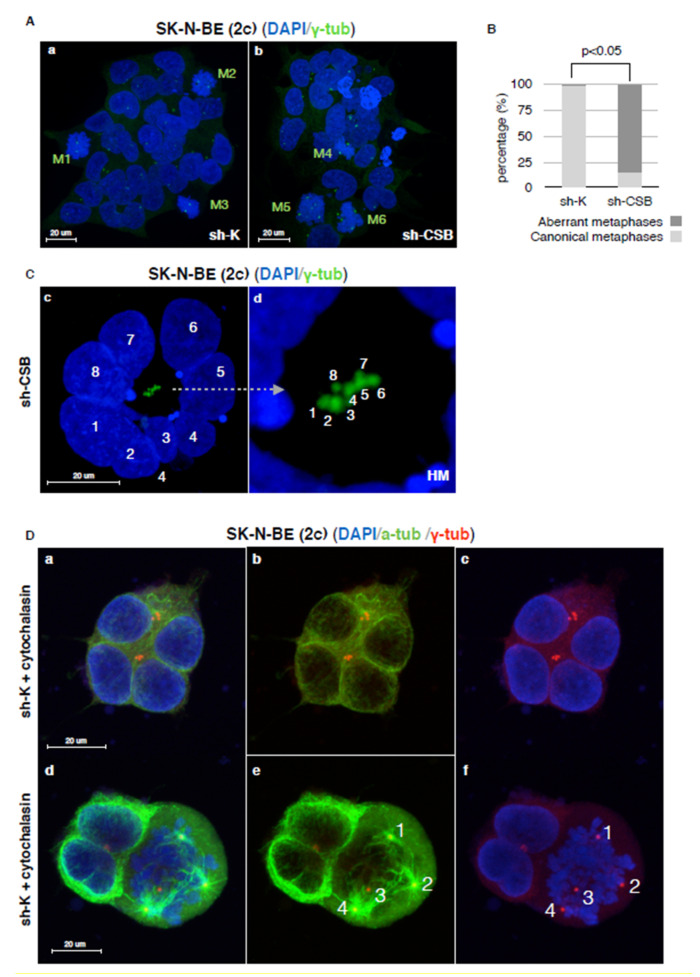
CSB suppression determines cytokinesis failure. Confocal micrographs of SK-N-BE (2c) sh-K (panel a), and sh-CSB (panel b) cells stained for DNA (blue) and γ-tubulin (green). Mitosis is indicated in green (**A**). Graph showing the percentage of canonical and aberrant metaphases in SK-N-BE (2c) sh-K and sh-CSB cells (n = 500 × 3 independent experiments) (**B**). Confocal micrographs of SK-N-BE (2c) sh-CSB cells stained for DNA (blue) and γ-tubulin (green). Nuclei are indicated by white numbers (panel c). Centrosomes are indicated by white numbers (panel d). HM indicates high magnification of relative indicated area (**C**). Confocal micrographs of SK-N-BE (2c) sh-K cells after cytochalasin treatment stained for DNA (blue), α-tubulin (α-tub, green), and γ-tubulin (γ-tub, red). Supernumerary centrosomes are indicated by white numbers (panels e and f) (**D**).

## Data Availability

Not applicable.

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
