# Peer review of "Neuroblastoma Cells Depend on CSB for Faithful Execution of Cytokinesis and Survival"

_ijms, 2021, doi:10.3390/ijms221810070_

Round 1
Reviewer 1 Report
I was not able to see the figures. I need those pictures in order to be able to review the paper. Minor typing mistakes are attached with the file

Author Response
We are sorry for the inconvenient which is not our fault since we loaded, as required by the journal policy the PDF of Figures. Said that, we are currently loading the PDF of manuscript and Figures at your convenience - now improved keeping into account the concerns of Reviewer #2

Reviewer 2 Report
In this manuscript the authors look further in to how the Cockayne syndrome group B (CSB) protein influences the survival of cancer cells, specifically neuroblastoma cells. Previous research has shown that the CSB protein is expressed to elevated levels in a panel of different cancer cell lines from various tissue types. These cells are specifically sensitive to depletion of CSB compared to normal cells. The authors extend these findings to look specifically in neuroblastoma cells and observe that there is a significant increase in CSB expression in two different cell types compared to normal neural cells. The authors then use one of the neuroblastoma cell lines and demonstrate that depletion of CSB leads to a significant growth defect. This growth defect is a result of an increased rate of apoptosis, centrosomal abnormalities and spindle multipolarity. While this manuscript provides useful insight into why some cancer cells may be specifically sensitive to CSB depletion the impact of these findings could be stronger.
Major comment:
- I only have one major comment. While the results are presented very clearly describing the effects of CSB depletion in SK-N-BE(2c) cells, the impact of this paper would be much stronger if some of the main experiments could also be completed for the normal neural cells (RenVM) and/or the other neuroblastoma cell line (IMR-32). This could be looking at apoptosis and centrosomal abnormalities/spindle multipolarity.
Minor comments:
- The level of cleaved caspase 3 shown by western in Figure 3F should be quantified as it is stated in the text that the experiment was already completed in triplicate.
- Beyond just representative images, the percentage of multinucleated cells and micronuclei per cell should be quantified in Figure 4A and cells with multipolarity in Figure 4B. This will further demonstrate how dramatic the defect is after CSB-depletion.
- Make sure all abbreviations are defined in the text. The abbreviations APR and CIN are never defined.
Author Response
Dear Reviewer#2,
we really thank your for the constructive concerns you raised that help us to considerably improve our work. Please find our answer point by point to your questions and in attachment PDF of the revised version of manuscript and Figures.
Reviewer #2: Major comment: I only have one major comment. While the results are presented very clearly describing the effects of CSB depletion in SK-N-BE(2c) cells, the impact of this paper would be much stronger if some of the main experiments could also be completed for the normal neural cells (RenVM) and/or the other neuroblastoma cell line (IMR-32). This could be looking at apoptosis and centrosomal abnormalities/spindle multipolarity.
Our Answer: As suggested by the Reviewer we introduced data concerning normal RenVm cells suppressed for CSB expression and we checked for apoptosis rate (Fig 3A panels c-d and Fig 3B), caspases activation (Fig.3C-I) and centrosomal abnormalities/spindle multipolarity (Fig.S1).
Reviewer #2: The level of cleaved caspase 3 shown by western in Figure 3F should be quantified as it is stated in the text that the experiment was already completed in triplicate.
Our Answer: As required we introduced a graph showing caspase 3 quantification (Fig. 3I).
Reviewer #2: Beyond just representative images, the percentage of multinucleated cells and micronuclei per cell should be quantified in Figure 4A and cells with multipolarity in Figure 4B. This will further demonstrate how dramatic the defect is after CSB-depletion.
Our Answer: We introduced in the text the quantifications required.
Reviewer #2: Make sure all abbreviations are defined in the text. The abbreviations APR and CIN are never defined.
Our Answer: We took into consideration the concern.

Round 2
Reviewer 2 Report
In this revised manuscript the authors have significantly increased the impact of their results by comparing the effect of CSB depletion between "normal" and neuroblastoma cells. I believe that this manuscript is suitable for publication currently.